# Current Status of Information and Communication Technologies Utilization, Education Needs, Mobile Health Literacy, and Self-Care Education Needs of a Population of Stroke Patients

**DOI:** 10.3390/healthcare13101183

**Published:** 2025-05-19

**Authors:** Mi-Kyoung Cho, Aro Han, Hyunjung Lee, Jiwoo Choi, Hyohjung Lee, Hana Kim

**Affiliations:** 1Department of Nursing Science, School of Medicine, Chungbuk National University, 1 Chungdae-ro, Seowon-gu, Cheongju 28644, Republic of Korea; ciamkcho@gmail.com (M.-K.C.); adodo23@cbnuh.or.kr (A.H.); jj4472jj@gmail.com (H.L.); cjw0681@cbnuh.or.kr (J.C.); dlgywjd6711@cbnuh.or.kr (H.L.); 2Department of Nursing Science, College of Life and Health Sciences, Hoseo University, Asan 31499, Republic of Korea

**Keywords:** stroke, self-care, health literacy, digital health, health knowledge, health communication, patient education as topic

## Abstract

Background/Objectives: With the rising prevalence of chronic diseases and an aging population, the incidence of stroke is continuously increasing, which leads to higher medical costs. Stroke carries a high risk of recurrence, necessitating ongoing self-care and lifestyle changes, for which education is crucial. The aim of this study is to identify the ICT utilization education needs, mobile health literacy, and self-care education needs of stroke patients and confirm the differences in mobile health literacy and self-care education needs according to ICT utilization to establish a basis for self-care intervention. Methods: The study included 100 stroke patients diagnosed at three general hospitals or higher in City C, hospitalized or visiting neurology and neurosurgery outpatient clinics. A survey was conducted from 7 July 2023 to 30 May 2024. The survey cites computers, the Internet, live broadcasting technology, recorded broadcasting technology, and telephony as examples of ICTs. The gathered data were analyzed using descriptive statistics, independent *t*-tests, one-way ANOVA, and the Pearson correlation coefficient. Results: The final analysis included 100 people, with 64 participants being men and an average age of 57.75 ± 12.30 years. Self-care education needs showed no significant differences based on general or disease-related characteristics. Many patients could use smart devices but experienced difficulties in searching for information. The main reasons for using smart devices included acquiring disease-related information and accessing resources without time limitations. The use of ICT services that provide disease-related information was low, 70% of participants were willing to use them in the future. Additionally, they preferred doctor-led education sessions once a month, lasting no longer than 30 min each. Mobile health literacy was significantly higher among those willing to use ICT services. Conclusions: Mobile health literacy was significantly higher in the group willing to use ICT services than in the group unwilling. Self-care education needs were both highly important and necessary in the group willing to utilize ICT, but no statistically significant difference was found.

## 1. Introduction

Stroke is a cerebrovascular disease and the fourth leading cause of death in Korea after malignant neoplasms, heart disease, and pneumonia [1]. More than 100,000 strokes occur each year in Korea [2]. This incidence rate is expected to increase because of the increase in chronic diseases and the aging population [2]. The number of stroke patients increased by 7.1%, from 591,000 in 2018 to 634,000 in 2022. The total medical expenses increased by 29% from approximately USD 1.4 billion to USD 1.8 billion during the same period. Hence, the number of patients and treatment costs have increased significantly [3].

Stroke is a dangerous disease with a high recurrence rate [4]. A meta-analysis by Lin et al. [4] reported a 10.4% stroke recurrence rate within one year over the past 10 years and 14.8% within 5 years. Such a high recurrence rate can place a tremendous financial and emotional burden on the patients and their families. Hence, continuous management is essential for preventing stroke recurrence. Na and Ryu [5] reported that improving lifestyle habits and implementing self-care are critical for preventing stroke recurrence. Hence, patients and medical staff should know this and pay more attention to appropriate prevention and management.

Stroke-related knowledge is the most critical factor in implementing self-care in stroke patients. Soto-Cámara et al. [6] stated that stroke patients had low levels of knowledge about the stroke warning signs and risk factors, and Wang et al. [7] reported that stroke-related knowledge was lacking. Therefore, active intervention is needed to resolve the lack of knowledge in stroke patients. Accordingly, accurately identifying the patient’s self-care education needs is essential.

Authors of several studies have examined the patients’ needs for self-care education. Shook and Stanton [8] reported that stroke patients have a high need for recovery and secondary prevention. Hafsteinsdóttir et al. [9] and Yonaty and Kitchie [10] reported that stroke patients have a strong need for self-care. Nevertheless, recent studies are insufficient, and despite some studies on caregivers of stroke patients in Korea, few studies have examined the educational needs of stroke patients.

Recently, due to the development of ICT (Information and Communication Technology), there have been increasing cases of stroke patients obtaining health information through channels such as smartphones, the Internet, and broadcast communication technology. Lee and Chang [11] reported that health information on stroke knowledge, warning symptoms, and treatment methods is provided through various media due to the development of ICT in modern society. Gustavsson et al. [12] stated that the use of ICT has a positive effect on the recovery of stroke patients in daily life, and Lee et al. [13] stated that ICT can contribute to the early recognition of health changes in stroke patients, which can contribute to health promotion and help reduce medical costs. These studies suggested that the use of ICT is essential for promoting self-care for stroke patients. Therefore, research is needed to identify the current status of ICT use by stroke patients, analyze its limitations, and seek ways to utilize ICT to promote self-care.

Mobile health literacy has a significant impact on the self-care abilities of stroke patients through ICT. Son and Song [14] reported that the self-management of cardiovascular disease patients, including stroke patients, varies according to their ability to utilize health information. Park et al. [15] reported that cultivating digital knowledge and skills is necessary because stroke patients are exposed to health information through smartphones and the Internet. Choi et al. [16] stated that people with higher mobile health literacy can collect reliable information and promote health promotion behaviors through ICT.

Hence, low mobile health literacy can adversely affect self-care promotion through ICT in stroke patients. Kumar et al. [17] and Vollbrecht et al. [18] also reported that low mobile health literacy can be a significant barrier to accessing health information. Mackert et al. [19] stated that low mobile health literacy restricts the use and limits the perception of the utility of digital health technology. The purpose of this study is to identify the utilization of ICT, education needs, mobile health literacy, and self-care education needs of stroke patients and confirm the differences in mobile health literacy and self-care education needs based on ICT utilization. It provides essential data for developing nursing interventions to promote the self-care of stroke patients. The research question of this study is, “Is there a difference in mobile health literacy and self-care education needs according to the utilization of ICT among stroke patients in Cheongju, South Korea?”

## 2. Materials and Methods

### 2.1. Study Design

This descriptive survey examined ICT utilization, education needs, mobile health literacy, and self-care education needs of stroke patients, confirming the differences in mobile health literacy and self-care education needs based on ICT utilization.

### 2.2. Study Population and Sampling

The participants were patients diagnosed with stroke at three general hospitals located in City C who were visiting the outpatient clinics of the neurology and neurosurgery departments or were hospitalized. The selection criteria for the participants were as follows: (1) adults aged 18 years or older, (2) patients diagnosed with stroke at least one month prior, (3) National Institutes of Health Stroke Scale (NIHSS) score of less than 10, (4) smartphone users, (5) those who could communicate and fill out the questionnaire, and (6) those who understood the purpose of the study and voluntarily agreed to participate. The exclusion criteria were those who (1) had difficulty understanding the questionnaire due to cognitive decline with an NIHSS score of 10 or higher and (2) had missing responses. Since an NIHSS score of 10 or higher indicates severe stroke symptoms, we excluded participants with a score of more than 10 as we judged that they may have difficulty understanding the questionnaire. The number of research participants required for this study was calculated using a sample size calculation program, the G*power program 3.1.9.7 (Heinrich-Heine-Universität Düsseldorf, NRW, Germany). According to a previous study [20], the number of participants was calculated with a two-sided test, a significance level of 0.05, an effect size of 0.25, a power of 0.80, and 12 predictor variables. The appropriate number of participants in the study was determined to be 81. The number of study participants was calculated to be 102, considering a 20% dropout rate.

### 2.3. Measurements

#### 2.3.1. Participants Characteristics

The participants’ general characteristics consisted of six items: age, sex, education, economic status, health status, and status of family care. Economic and health status were measured using a 10-point scale, where one indicated the lowest and ten indicated the highest levels. Disease-related characteristics were assessed based on stroke type, time since stroke onset (in years), and the number of comorbidities, including hypertension, diabetes, hyperlipidemia, and heart disease.

#### 2.3.2. ICT Utilization and Education Needs

ICT refers to various technological tools and resources to transmit, store, create, share, and exchange information. These technological tools and resources include computers, the Internet (websites, blogs, and email), live broadcasting technology (radio, television, and webcasting), recorded broadcasting technology (podcasting, audio and video players, and storage devices), and telephony (landline or mobile devices, satellite, and visio/video conferencing). In the present study [21], after revising and supplementing a previous study, a team of three experts (one member of the medical information management team, one professor of adult nursing, and one professor of informatics) was formed to measure the content validity before use. The content validity was measured using the I-CVI (Item-level content validity index) and evaluated on a one to four scale [22]. The content validity was calculated by assigning one point to each item that experts evaluated as three or four points and zero points to each item that experts evaluated as one or two points. The content validity is considered high when it is 0.8 or higher when evaluated by three or more experts. The average I-CVI for the status of ICT utilization and the need for ICT utilization education was 0.8 or higher, suggesting that the validity of measuring the status of ICT utilization and the need for education was considered high. The five items on the status of ICT utilization are as follows: level of smart device usage, whether and why smart devices are used, time of smart device usage, route for obtaining disease-related information, and whether disease-related services are being used. The four items on the need for ICT utilization education were as follows: willingness to use ICT education services, desired method of providing health or social services to the participants, information on stroke and the treatment that the participant would like to receive, and the subject, time, and frequency of education on stroke disease and treatment, for a total of nine items.

#### 2.3.3. Mobile Health Literacy

Mobile health literacy refers to obtaining, processing, and understanding health information and services using portable, wireless devices to identify or solve health problems [23]. The mobile health literacy of stroke patients was measured using the eHealth Literacy Scale (eHEALS) tool developed by Norman and Skinner [24] and modified and adapted by Chang et al. [25]. This tool consists of ten questions. The first two questions were additional to determine the interest in using Internet health information when making health-related decisions but were not included in the scoring process. Eight questions (3–10) reflecting Internet health information literacy were scored on a five-point Likert scale, with one point being ‘not at all’ and five points being ‘very much’. The total score ranged from eight to forty, with a higher score indicating higher mobile health literacy. The reliability of the tool was Cronbach’s α = 0.88 in the study of Norman and Skinner [24], Cronbach’s α = 0.88 in the study of Chang et al. [25], and Cronbach’s α = 0.96 in this study.

#### 2.3.4. Self-Care Education Needs

Education needs are the subjective desire to acquire knowledge or skills related to health issues, disease prevention, and health promotion [26]. The self-care education needs measurement tool used in this study was developed based on a literature review and the researcher’s clinical experience, and five experts verified the content validity before use (two neurologists, two nurses with more than five years of clinical experience in stroke intensive care units, and one nursing professor). The content validity was measured by calculating the item level content validity index (I-CVI), and the score was calculated similarly to ICT utilization and education needs [22]. The average I-CVI for self-care education needs was >0.8, indicating the high validity of self-care education needs. The self-care education needs of this tool consisted of importance and necessity, and the sub-domains were composed of nine questions on disease definition: three questions on complications and prognosis, three questions on social support, and three questions on lifestyle management, for a total of thirty-six questions with eighteen questions each. The items were measured on a five-point Likert scale, from one point for ‘not at all’ to five points for ‘very much’. The total score was calculated by summing the scores for each item. The score ranged from 18 to 90 points, with a higher score indicating higher importance and necessity of self-care education. The tool’s reliability was Cronbach’s alpha = 0.96 for the importance of stroke education and Cronbach’s alpha = 0.96 for the necessity of stroke education.

### 2.4. Data Collection and Ethical Considerations

Data were collected using a self-reporting questionnaire from subjects who voluntarily consented after the researcher explained the purpose and content of the study. The questionnaires were collected directly by the researcher after being sealed in a return envelope to protect the participant’s personal information and collected directly by the researcher. Participants who responded to the questionnaire were given a mouthwash worth about USD 2 as a token of gratitude. The study purpose and the expected benefits and risks of participating were explained to the participants who met the selection criteria. The anonymity and confidentiality were described, and signed written consent forms were obtained. The subjects were informed that they could refuse to respond to the survey if they did not wish to participate. In addition, they could stop participating at any time during the study without any disadvantage. No subject expressed a desire to stop participating during the study.

### 2.5. Data Analysis

The data collected in this study were analyzed using the SPSS 29.0/Windows program (IBM Corp., Armonk, NY, USA). Descriptive statistical analysis was conducted on the participants’ general characteristics, disease-related characteristics, and significant variables. In addition, the differences in mobile health literacy and self-care education needs according to the willingness to use ICT education services were analyzed using an independent *t*-test. The differences in self-care education needs according to participant characteristics, ICT utilization, and education needs were also analyzed using an independent *t*-test and one-way ANOVA. The Pearson correlation coefficient was used to assess the correlation between the importance and necessity of mobile health literacy and self-care education needs.

## 3. Results

This study was conducted on 100 outpatients of neurology and neurosurgery departments at three general hospitals or higher in City C from 7 July 2023 to 30 May 2024, and received approval from the Institutional Review Board of C University Hospital before data collection (IRB No. 2023-06-027-001). The final analysis included the survey results of 100 of the 102 participants who participated in the study, excluding 2 respondents who gave insincere responses.

### 3.1. Participants’ Characteristics

The participants’ characteristics are as follows: average age of 57.76 ± 12.30 years (range: 20–85 years), 64 men (64.0%), 72 (72.0%) with ischemic stroke, economic status of 5.01 ± 2.17, and health status of 5.35 ± 2.08. The average time since stroke onset was 3.06 ± 4.44 years, and the average number of comorbidities was 2.00 ± 0.82 (Table 1).

### 3.2. Differences in Self-Care Education Needs According to the Participants’ Characteristics

This study examined whether there were differences in the importance and necessity, which are sub-domains of the need for self-care education, according to the participants’ general characteristics. The need for self-care education was higher in the group younger than the average age (average 57.76 ± 12.30 years) and the group diagnosed with ischemic stroke rather than hemorrhagic stroke. In addition, the need for self-care education was higher in the group diagnosed with middle school education or lower, those without a caregiver, with an above-average economic status, and with a below-average subjective health status. The need for self-care education was higher in the group with a shorter-than-average disease time since stroke onset (average 3.06 ± 4.44) and without complications. However, no statistically significant difference in each characteristic was observed between the groups. The need for self-care education was higher in the group with a shorter-than-average time since stroke onset (average: 3.06 ± 4.44 years) and no complications.

### 3.3. Result of ICT Utilization and Education Needs Measurement

Regarding the ICT utilization of the participants, the largest number of participants (39.0%) knew how to use smart devices to some extent but had difficulty finding the information they wanted, and 78 participants (78.0%) used smart devices. The reasons for using smart devices were the ease of obtaining disease-related information for 27 participants (34.6%) and the ease of obtaining information regardless of time for 21 participants (26.9%). The most common reason for not using smart devices was a lack of skill in operating the devices for 18 participants (81.8%). Sixty-eight participants (68.0%) were not currently using any services related to their disease.

Regarding the need for education on ICT utilization, 70 people (70.0%) were willing to use ICT when providing health services. The most common method for receiving disease-related information was text or KakaoTalk messenger with 59 people (59.0%). The information requested was the treatment methods, self-care methods, and disease-related information by 46 (46.0%), 29 (29.0%), and 23 (23.0%) people, respectively. The most preferred education was by a doctor (n = 64, 64.0%). The most common education period and education time were once a month (n = 55, 55.0%) and within 30 min, respectively (n = 68, 68.0%) (Table 2).

### 3.4. Differences in the Need for Mobile Health Literacy and Self-Care Education According to the Intention to Use ICT Services

The total scores of the participants’ mobile health literacy, the importance of self-care education needs, and the necessity of self-care education needs were 23.49 ± 7.93, 78.69 ± 10.71, and 78.26 ± 10.49, respectively. In the sub-domain of the importance of self-care education needs, disease definition, complications and prognosis, and lifestyle management scored 39.88 ± 5.43, 13.19 ± 1.96, and 12.54 ± 2.30, respectively. In the sub-domain of the necessity of self-care education needs, the disease definition scored high at 39.80 ± 5.20. The mobile health literacy of the group willing to use ICT services was significantly higher than that of the group unwilling (t = 4.11, *p* < 0.001), and the necessity of self-care education was similar (Table 3).

## 4. Discussion

This study identifies the ICT utilization, education needs, mobile health literacy, and self-care education needs of stroke patients and confirms the differences in mobile health literacy and self-care education needs according to ICT utilization. Although few participants were currently using ICT services related to their diseases, 70% said they would like to use them in the future. In addition, they wanted to receive disease-related information using familiar methods, preferred doctors as educators, and preferred education once a month for 30 min per session. In addition, mobile health literacy was significantly higher in the group willing to use ICT services than in the group unwilling. Self-care education needs were both highly important and necessary in the group willing to utilize ICT, but no statistically significant difference was found.

The average age of the study participants was 57.76 years, and there were more males (n = 64, 64%) than females (n = 36, 36%). Previous studies also confirmed that the proportion of male stroke patients was more than half [27], with reported figures of 53.7% [28], 66.25% [29]. Moreover, domestic statistics also confirmed that the proportion of male patients was 59.8% [1]. Therefore, the results appropriately show the current situation of stroke patients. This is because the stroke incidence rate is higher in men than in women, and men may be more susceptible to stroke due to genetic polymorphisms [30]. In addition, the proportion of smoking and diabetes, risk factors for stroke, is higher in men than in women [31]. Furthermore, the age of stroke onset in women is higher than in men, and health management skills are lower [31]. Among the stroke classifications, ischemic stroke was more common at 72% than hemorrhagic stroke. This result is consistent with previous studies [32,33], showing that ischemic stroke accounts for approximately 85–87% of all strokes. This result is because ischemic stroke has various causes, such as thrombosis, atherosclerosis, cardioembolism, and small vessel occlusion [32]. The average time since stroke onset among the study participants was 3.06 years, suggesting that stroke patients have a high risk of recurrence. They undergo continuous rehabilitation and management even after the acute phase and live with various after-effects for many years [31,34]. Therefore, patients in the relatively early stage of the disease participated. This result is consistent with previous studies showing that chronic disease patients tend to try various interventions in the early years and show an active attitude toward disease recovery [35].

The average number of comorbidities in the participants of this study was two. This result is similar to the survey by She Rui [36], in which only 9%, 29.6%, 30.1%, and 31.3% of patients had zero, one, two, and three or more comorbidities, respectively. The most common comorbidities in stroke patients were hypertension (74.7%), diabetes (28.5%), dyslipidemia (23.3%), coronary artery disease (19.1%), and obesity (17.7%) [31,36]. These comorbidities have been identified as risk factors for stroke, which adversely affect the prognosis and treatment outcomes of stroke patients and require appropriate management [31].

An analysis of the difference in self-care education needs according to the general characteristics and disease-related characteristics of the subjects showed higher self-care education needs in the following groups: the group under 57.76 years of age, female, diagnosed with ischemic stroke, middle school education or lower, no caregiver, above average economic status, below average subjective health status, short disease duration, and no complications. Nevertheless, the differences were not statistically significant because stroke patients require continuous rehabilitation and treatment to prevent recurrence even after recovery from the acute phase [31]. In addition, the period was short at 3.06 years, and it was presumed that the study participants were still receiving medical help by undergoing outpatient treatment periodically. In light of the results showing that patients with a short disease duration have a high need for education [37,38], the level of need by the study participants was not low, but they were patients in the early stage with similar educational needs overall and showed no difference according to the disease duration. In contrast to the results of the present study, many studies have confirmed that women are more active in health management [39,40] and have a higher need for self-care education. In addition, a lower education level indicated a higher tendency for education needs [41,42]. In this study, 77% of the participants had a high school diploma or higher. Hence, the education level was similar to other studies. The need for self-care education was somewhat higher in cases without complications because pneumonia and heart failure, which are major complications of stroke, require active treatment due to the nature of the diseases [43]. Therefore, in cases of these diseases, it is possible to obtain disease-related information continuously through outpatient visits or hospitalization. Thus, it can be assumed that the need for additional education has decreased.

These results also confirmed that stroke patients do not show consistent characteristics according to general or disease-related characteristics in terms of self-care education needs. Finch et al. [44] confirmed that stroke survivors have diverse needs regarding education methods, timing, content, and format. Therefore, health services tailored to individuals should be provided flexibly according to the patient’s or family’s living situation and the disease severity [45]. Implementing stroke education tailored to the individual’s needs can meet the needs of stroke survivors and caregivers from hospitals to local communities [44].

The ICT utilization status of the study participants showed that many knew how to use smart devices but had difficulty finding the information they wanted. This is presumed to be the result of decreased cognitive ability with age or decreased ability to learn and understand online health information because of cognitive decline caused by stroke [46]. In addition, the generation gap between digital natives and immigrants might play a role [47]. On the other hand, smart devices are used by 78% of the population. Therefore, providing interventions to improve digital literacy will contribute to the spread of digital health service use for self-care. Many people said that the reason for not using smart devices was that they were not skilled at operating the devices. This result is consistent with Weijia et al. [48], who stated that people are reluctant to use them owing to the fear of system damage because it is difficult to respond appropriately in the event of a network failure. Therefore, it is necessary to provide real-time user guide support methods, such as providing FAQs and chatbots, to ensure smooth service participation when providing digital health services [49]. The use of ICT is increasing, and mobile apps and online education can be effective ways for patients to receive continuous education and management without visiting an outpatient clinic. Vitali et al. [50] used ICT technology to improve motor skills after a stroke. Furthermore, a survey of remote rehabilitation using ICT by Yanghui Xing et al. [51] was effective for cognitive impairment, speech and language disorders, and depression reduction. Therefore, it is essential to identify and supplement the level of use of ICT devices and the areas where stroke patients find difficulty, as well as strengthen patient education and support so that stroke patients can use ICT devices efficiently.

Despite the advantages of ICT technology, 68% of the participants were not currently using the service. This may be because, although many stroke-related apps have been released domestically when looking at mobile health apps among ICT technologies, most of them only provide limited information such as disease information, prevention information, surgical methods, or drug treatments [52]. Even in the case of apps that provide rehabilitation or exercise information, most apps provide pictures and videos of how to perform them [52]. Seventy percent of the participants responded that they were willing to use ICT services when provided, which was higher in the present study than in the studies by Shibuta et al. [53] (50% for diabetic patients) and Lin et al. [54] (47% for coronary artery disease patients). Therefore, stroke patients feel the need for better ICT services, which supports the present results showing a significant correlation between the willingness to use ICT services and mobile health literacy in that a higher mobile health literacy means a higher understanding of ICT services and a lower entry barrier for using various services.

When providing disease-related information, many participants wanted to receive information through familiar means, such as text messages or KakaoTalk, and the information they wanted was mostly treatment methods, self-care methods, and disease-related information. Stroke patients may have difficulty using smart devices proficiently because of functional disabilities, such as cognitive decline and hemiplegia [21]. Moreover, the information they wanted to receive first differed from that of the residents [21]. According to Kim et al. [55], the awareness rate of stroke symptoms and risk factors in Korea was significantly low at 50–60%. In addition, Hong et al. [56] reported that less than 50% knew the definition and the risk factors of stroke. Hence, education about strokes is insufficient, highlighting the urgent need for stroke education. Stroke is a disease in which the golden time is essential. Moreover, effective prevention and treatment may be difficult if awareness is insufficient for the public and stroke patients because the recurrence rate and subsequent disability rate are high. Furthermore, individual education on recurrence prevention, counseling, emotional support, and encouragement for individual risk factors is important for stroke patients to recover quickly and prevent recurrence [57]. Therefore, helping patients accurately understand their illness and practice active self-management through individual education is important.

Among the study participants, 64% wanted doctors as educators because doctors have a scientific and professional status in the relationship between patients and medical professionals [58]. Moreover, the doctor’s recommendations are considered essential to patients and their families [58]. Many participants preferred the education period to be once a month and the education time to be 30 min. Patients prefer short information delivery over extended education and make more effective judgments [59]. Forty to eighty percent of patients do not accurately remember the education content after consulting with a doctor, suggesting that complex or detailed information may be ineffective for patients [59,60].

The mobile health literacy was high in the group willing to use ICT services. The score of mobile health literacy was 58.75 out of 100, indicating an intermediate level of ability. This score was lower than the 76.72 points measured by Chang et al. [25] and the 67.95 points measured by Lee et al. [61], using the same tool for the general public. This result is interpreted as a relatively low score compared to the general public because the participants of this study were stroke patients, a vulnerable group. Nevertheless, the result is consistent with previous research showing that mobile health literacy is low among chronic disease patients [62,63]. The fact that mobile health literacy was higher in the group that was willing to use ICT services is consistent with previous studies that showed that people with high eHealth literacy tend to use mHealth apps more [64]. Nevertheless, this may lead to digital disconnection and health inequality in the case of vulnerable groups with limited access to mobile services and the Internet [65]. Petretto et al. [66] addressed this by suggesting simplifying complex interfaces and information flows, using intermediaries, and developing mechanisms for people at risk of disadvantage in accessing digital health to provide helpful input in designing and implementing telemedicine interventions. Sieck et al. [67] suggested recognizing the level of access to devices and internet connectivity in the community and establishing initial service use and ongoing technical support for patients. They also suggested ways to improve health literacy by providing interactive educational content, symptom tracking, medication reminders, and peer support communities in mobile service apps [68]. This will enhance digital service accessibility, increase the participation of vulnerable groups, and ultimately help achieve health equity.

In the group willing to use ICT services, both the importance and necessity of self-care education were high, but no statistically significant difference was confirmed. This result was because the group willing to use the service, had a more proactive attitude, and was more interested in self-care education than those who were unwilling. This result is consistent with the study by Ifejika et al. [69], which found that people with active physical activity were more likely to use health apps. Nevertheless, the period of illness of the subjects who participated in this study was not long, so no significant difference was confirmed. In addition, it should be considered that stroke affects cognitive domains and that dementia develops in about 30% of patients within a year, which may affect the use of ICT services [69].

This study had several limitations. First, the three hospitals that recruited study participants were located in one city, which limits the generalizability of the results. ICT can vary depending on the level of economic development, infrastructure, and accessibility to medical care in a region. Therefore, additional research targeting various regions and hospitals is needed. Second, the average age of the participants in this study was lower than that of stroke patients, which limits the generalization of the study results to all stroke patients. Third, the cross-sectional study design limited the ability to identify causal relationships. Future studies will analyze changes over time through longitudinal studies. Fourth, although this study treated ischemic stroke and hemorrhagic stroke patients together, the characteristics of each group should be investigated separately in future studies. Finally, as a policy recommendation, it is necessary to develop and distribute ICT education programs targeting stroke patients so that patients can use the technology more effectively. In addition, these programs must be made more accessible to patients and provide customized education tailored to individual patients’ needs to improve their self-management abilities.

## 5. Conclusions

In this study, people living with stroke from Korea with an average age of approximately 58 provided evidence that they utilize health information effectively through ICT and have a strong demand for self-care education. Regardless of the various demographic backgrounds and ICT utilization levels, all stroke patients recognized the need for self-care education. These results highlight the need for comprehensive and accessible self-care education programs. Stroke patients must be supported in performing self-care effectively. Future studies should design programs that can be applied effectively to all patients and develop patient-tailored ICT education programs and support systems to improve the quality of life of stroke patients.

## Figures and Tables

**Table 1 healthcare-13-01183-t001:** Differences in Self-care Educational Needs according to Characteristics of Participants (N = 100).

Characteristics	Categories	n (%)	M ± SD (Min–Max)	Educational Needs			
				Importance		Necessity	
				M ± SD	t or F (*p*) Scheffé	M ± SD	t or F (*p*) Scheffé
Age (year)	<57.76	39 (39.0)	57.76 ± 12.30 (20–85)	79.61 ± 10.41	0.69 (0.493)	78.92 ± 10.57	0.50 (0.616)
	≥57.76	61 (61.0)		78.10 ± 10.94		77.83 ± 10.50	
Sex	Male	64 (64.0)		77.39 ± 11.03	−1.63 (0.106)	76.92 ± 10.75	−1.72 (0.089)
	Female	36 (36.0)		81.00 ± 9.98		80.63 ± 9.70	
Types of stroke	Ischemic	72 (72.0)		79.54 ± 10.69	1.28 (0.204)	78.84 ± 10.75	0.90 (0.372)
	Hemorrhagic	28 (28.0)		76.50 ± 10.65		76.75 ± 9.81	
Education	≤Middle school	23 (23.0)		79.69 ±9.22	0.14 (0.874)	79.00 ± 8.44	0.08 (0.928)
	High school	49 (49.0)		78.49 ± 11.13		78.10 ± 10.55	
	≥College	28 (28.0)		78.21 ± 11.42		77.92 ± 12.12	
Status of family care	Family	80 (80.0)		78.53 ± 10.66	0.33 (0.721)	78.05 ± 10.40	0.37 (0.689)
	Non-family	6 (6.0)		76.50 ± 13.57		76.33 ± 13.90	
	None	14 (14.0)		80.50 ± 10.33		80.29 ± 10.02	
Economic status	<5.01	64 (64.0)	5.01 ± 2.17 (0~10)	78.39 ± 10.86	−0.37 (0.712)	77.97 ± 10.85	−0.37 (0.713)
	≥5.01	36 (36.0)		79.22 ± 10.58		78.78 ± 9.96	
Health status	<5.35	58 (58.0)	5.35 ± 2.08 (0~10)	79.65 ± 10.72	1.06 (0.292)	78.97 ± 10.69	0.79 (0.432)
	≥5.35	42 (42.0)		77.35 ± 10.72		77.29 ± 10.26	
Time since stroke onset (years)	<3.06	71 (71.0)	3.06 ± 4.44 (0.17~26.17)	79.29 ± 10.63	0.88 (0.379)	79.20 ± 10.11	1.40 (0.163)
	≥3.06	29 (29.0)		77.20 ± 10.95		76.0 ± 11.2	
Number of Comorbidity	None	33 (33.0)	2.00 ± 0.82 (0~4)	80.87 ± 10.02	1.05 (0.353)	80.33 ± 9.90	1.04 (0.357)
	1	34 (34.0)		77.32 ± 11.84		76.73 ± 11.63	
	≥2	33 (33.0)		77.90 ± 10.13		77.76 ± 9.79	

M, mean; SD, standard deviation; Min, minimum; Max, maximum.

**Table 2 healthcare-13-01183-t002:** ICT utilization and Educational Needs (N=100).

Variables	Categories	n (%)	M ± SD	Min~Max
ICT utilization				
Usage of smart devices	No	22 (22.0)		
	Yes	78 (78.0)		
Level of smart device usage	Uses it very proficiently	20 (20.0)		
	Has a good level of proficiency and has no difficulty finding the desired information	27 (27.0)		
	Has some proficiency but struggles to find the desired information	39 (39.0)		
	Does not know how to use it at all	14 (14.0)		
Reasons for using smart devices	Ease of obtaining information related to disease	27 (34.6)		
	Improved understanding through educational materials such as videos	12 (15.4)		
	Ease of obtaining desired information by utilizing available time	21 (26.9)		
	Enables interaction with many people	16 (20.5)		
	Other	2 (2.6)		
Reasons for not using smart devices	Lack of proficiency in operating the device	18 (81.8)		
	Spending too much time finding the desired information	2 (9.1)		
	Does not feel the need	2 (9.1)		
Smart device usage Time (hours)	<3.06	61 (61.0)	3.06 ± 2.70	0~17
	≥3.06	39 (39.0)		
Sources of information	ICT Sources (TV, SNS, Youtube, Internet)	83 (83.0)		
	Non-ICT Sources (Family, Healthcare providers, Books)	17 (17.0)		
Availability of services used for disease-related information	No	68 (68.0)		
	Yes	32 (32.0)		
Educational needs for ICT utilization				
Willingness to use ICT services	Yes	70 (70.0)		
	No	30 (30.0)		
Preferred method of receiving Disease-related information	Text message/KakaoTalk messenger	59 (59.0)		
	Internet/Cafe/Band/Website	24 (24.0)		
	In-person/Booklet/Brochure	15 (15.0)		
	Other	2 (2.0)		
Information desired to be provided	Treatment Options After Diagnosis	46 (46.0)		
	Self-Care methods	29 (29.0)		
	Disease-related Information Other	23 (23.0) 2 (2.0)		
Preferred educators	Doctor	64 (64.0)		
	Nurse	18 (18.0)		
	External educators	11 (11.0)		
	Other	7 (7.0)		
Preferred duration of education	Once a week	28 (28.0)		
	Once a month	55 (55.0)		
	Once a year	17 (17.0)		
Preferred time for education	Within 30 min	68 (68.0)		
	One hour	28 (28.0)		
	Two hours	2 (2.0)		
	Other	2 (2.0)		

M, mean; SD, standard deviation; Min, minimum; Max, maximum; ICT, Information and Communications Technologies.

**Table 3 healthcare-13-01183-t003:** Differences in Mobile health literacy, Demand for Self-care education, and Intention to use ICT services (N = 100).

Variables	Items	M ± SD	Range	Scale Standardized Score		Intention to Use ICT Services		t (*p*)
						Yes (N = 70)	No (N = 30)	
				M ± SD	Range	M ± SD	M ± SD	
Mobile health literacy	8	23.49 ± 7.93	8~40	2.94 ± 0.99	1-5	25.47 ± 7.32	18.87 ± 7.46	4.11 (<0.001)
Educational needs								
Importance	18	78.69 ± 10.71	50~90	4.37 ± 0.60	2.8~5.0	79.60 ± 10.43	76.57 ± 11.26	1.30 (0.196)
Definition of disease	9	39.88 ± 5.43	26~45	4.43 ± 0.58	2.9~5.0	40.36 ± 5.20	38.77 ± 5.87	1.35 (0.181)
Complications and prognosis	3	13.19 ± 1.96	9~15	4.39 ± 0.65	3.0~5.0	13.31 ± 1.95	12.90 ± 1.99	0.97 (0.336)
Lifestyle management	3	12.54 ± 2.30	7~15	4.18 ± 0.76	2.3~5.0	12.81 ± 2.26	11.90 ± 2.29	1.85 (0.068)
Social support	3	13.08 ± 2.07	8~15	4.36 ± 0.69	2.7~5.0	13.11 ± 2.02	13.00 ± 2.23	0.25 (0.802)
Necessity	18	78.26 ± 10.49	50~90	4.35 ± 0.58	2.8~5.0	79.20 ± 10.65	76.07 ± 10.65	1.37 (0.172)
Definition of disease	9	39.80 ± 5.20	26~45	4.42 ± 0.58	2.9~5.0	40.34 ± 4.91	38.53 ± 5.69	1.61 (0.111)
Complications and prognosis	3	13.01 ± 1.93	9~15	4.34 ± 0.64	3.0~5.0	13.14 ± 1.94	12.70 ± 1.91	1.05 (0.295)
Lifestyle management	3	12.47 ± 2.19	7~15	4.16 ± 0.76	2.3~5.0	12.76 ± 2.34	11.80 ± 2.06	1.94 (0.055)
Social Support	3	12.98 ± 2.06	8~15	4.33 ± 0.69	2.7~5.0	12.96 ± 2.07	13.03 ± 2.08	−0.17 (0.867)

M, mean; SD, standard deviation; ICT, Information and Communications Technology.

## Data Availability

The original contributions presented in the study are included in the article, further inquiries can be directed to the corresponding authors.

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
