# Peer review of "Current Status of Information and Communication Technologies Utilization, Education Needs, Mobile Health Literacy, and Self-Care Education Needs of a Population of Stroke Patients"

_healthcare, 2025, doi:10.3390/healthcare13101183_

Round 1
Reviewer 1 Report
Comments and Suggestions for Authors
The article is well-written.
Abstract: I have one or two feedbacks on the abstract. The first one is, “Although ICT services were low,” and I wonder what kind of ICT services you are referring to and if the numbers are low. I think the number of services related to stroke is medium, but perhaps it relates to my geographical location.
Introduction: The introduction is good, and some details need improvement. You use either the word stated or reported after references. Look for other synonyms to make the text better.
I also encourage you to ask a research question after the aim. Could it include the geographical location you are investigating?
Materials and methods: You are referring to the type of stroke, but isn’t it more interesting to present the stroke patients’ type of impairment, such as speech, motoric, or cognitive? I assume that affects the usage of ICT more than the type of stroke.
What does the economic status of 5.01 ± 2.17 mean? You are in an international arena and must explain that to the reader. The same goes for health status.
You are presenting the usage of smart devices as the second question, even if it is a yes/no question. Is it so the respondents could move on even if they answered no to that question?
You are presenting social support twice in Table 3; how come?
Discussion: Parts of the discussion are more of an analysis, such as economic status. It is not used for further discussion, merely as information to the reader. I assume that you can divide the discussion into two parts, where one is information, and the other one is discussion, answering the aim/research question.
Author Response
Thank you for your careful review of this study. I will submit my response as a file.

Reviewer 2 Report
Comments and Suggestions for Authors
Thank you for the opportunity to review this manuscript. I left my suggestions and questions.
TITLE
“Survey on the current status of ICT utilization and education needs of stroke patients, mobile health literacy, and self-care education needs” OR “Current status of ICT utilization, education needs, mobile health literacy, and self-care education needs of a population of stroke patients” Please, think about this.
OBJECTIVES
Lines 14-16; “This study evaluates stroke patients' use of ICT, educational needs, and self-care education requirements related to mobile health literacy to establish a basis for self-care intervention.” / Lines: 82-85. “Therefore, this study examined the status of ICT utilization, educational needs, mobile health literacy, and self-care education needs of stroke patients, providing basic data for developing nursing interventions to promote the self-care of stroke patients.” / Lines 254-255: “This study examined the differences in the need for self-care education and mobile health literacy and self-care education according to the willingness to use ICT services.”
Please, rewrite the objectives.
ABSTRACT
-Objective Lines 14-16; “This study evaluates stroke patients' use of ICT, educational needs, and self-care education requirements related to mobile health literacy to establish a basis for self-care intervention.” / Conclusion Lines 26-28: “Stroke patients demonstrated a strong demand for self-care education and effectively utilized ICT for health information. Therefore, tailored ICT education programs may enhance their quality of life.” Please, answer the objectives in “Conclusion” section.
-Methods: Line 16: ”The study included 100 stroke patients…” Please, include information about the total number of responses (100) and summarize sociodemographic data of participants (age, gender) in “Results” section.
- Conclusion Lines 26-28: “Stroke patients … effectively utilized ICT for health information…” OR “In this study, stroke patients (average age of 57.76) … effectively utilized ICT for health information…” Please, think about this and correct the information.
INTRODUCTION
-Line 37-39: “The total medical expenses increased by 29% from KRW 1.8953 trillion to KRW 2.4457 trillion during the same period. Hence, the number of patients and treatment costs have increased significantly [3].” Please, the authors could also include the total medical expenses in dollars or euros.
-Lines 60-62: “Recently, the number of cases in which stroke patients obtain health information through ICT has increased because of the development of Information and Communication Technologies (ICT)” OR “Recently, the number of cases in which stroke patients obtain health information through Information and Communication Technologies (ICT) has increased because of the development of ICT”
MATERIALS AND METHODS
-Lines 93-94: “The selection criteria for the participants were as follows: 1) adults aged 19 years or older…” Why not adults aged 18 years or older? Please, explain this.
-Lines 94-95: “… patients diagnosed with stroke who had passed more than one month” Why? Please, rewrite this information.
-Lines 95-96: “…National Institutes of Health Stroke Scale (NIHSS) score of 10 or higher…” Why? Please, explain this in the text.
-Lines 98-99: “The exclusion criteria were those who 1) had difficulty understanding the questionnaire due to cognitive decline with an NIHSS score of less than 10…” “Less than 10”, is this correct?
-Lines 105-107: “One hundred and two subjects were selected, considering a dropout rate of 20%. Of the 102 questionnaires, 100 were used for the analysis after excluding the inadequate responses.” Please, this information could be moved to “Results” section.
-Lines 111-113: “The economic status and health status were measured on a 10-point scale—the disease-related characteristics comprised stroke types, duration (years), and the number of comorbidities.” This is confusing. Please, rewrite this.
-Line 113. “…the number of comorbidities.” - Please, explain in the text which comorbidities were considered.
-Line 172: This study was conducted on 100 outpatients…” Please, the total number of participants could be described in “Results” section.
-Lines 178-179: “The participants received 3,000 won worth of gargle solution as a token of gratitude for responding to the questionnaire”. I did not understand this information.
-The questionnaire could be available as supplementary material.
RESULTS
--Table 1: “Duration of stroke (year)” OR How long ago did the participant have a stroke? Please, rewrite the information.
-Line 201 “…64 men (64.0%)…” However, we can see in the table 1 : “Male 36 (36.0)”. Please, correct the information.
-Lines 201-202: “…economic status of 201 5.01 ± 2.17, and health status of 5.35 ± 2.08…” Please, explain this.
-Lines 213-215: “…those without a caregiver, with an above-average economic status, and with a below-average subjective health status.” Please, explain in the text “above-average economic status” and “below-average subjective health status”.
-Lines 220-221: “Regarding the ICT utilization of the participants, the largest number of participants (39.0%) knew how to use smart devices to some extent…” Could us consider 39% as the largest number of participants? Please, think about this.
-Lines 221-226: “…78 participants (78.0%) used smart devices. The reasons for using smart devices were the ease of obtaining disease-related information for 27 participants (34.6%) and the ease of obtaining information regardless of time for 21 participants (26.9%). The most common reason for not using smart devices was a lack of skill in operating the devices for 18 participants (81.8%).” OR “…78/100 participants (78.0%) used smart devices. The reasons for using smart devices were the ease of obtaining disease-related information for 27/78 participants (34.6%) and the ease of obtaining information regardless of time for 21/78 participants (26.9%). The most common reason for not using smart devices was a lack of skill in operating the devices for 18/22 participants (81.8%).” Please, think about this.
-Table 2. Educational needs for ICT utilization: Preferred method of receiving Disease-related information: 59+23+15+2 = 99 (not 100). Please, rewrite the information.
DISCUSSION
Another limitation of the study would be the fact that the participants were not mostly elderly (Participant average age of 57.76). Therefore, the results should be considered with caution.
CONCLUSION
Line 427-428. “Stroke patients utilize health information effectively through ICT and have a strong demand for self-care education” OR In this study, stroke patients utilize health information effectively through ICT and have a strong demand for self-care education
Author Response

(The authors gave the same response as above.)

Round 2
Reviewer 2 Report
Comments and Suggestions for Authors
Congratulations on the text. I have two questions.
1-NIHSS
Lines 105-109. “The exclusion criteria were those who 1) had difficulty understanding the questionnaire due to cognitive decline with an NIHSS score of 10 or higher and 2) had missing responses. Since an NIHSS score of 10 or higher indicates severe stroke symptoms, we excluded participants with a score of less than 10 as we judged that they may have difficulty understanding the questionnaire. “ OR “The exclusion criteria were those who 1) had difficulty understanding the questionnaire due to cognitive decline with an NIHSS score of 10 or higher and 2) had missing responses. Since an NIHSS score of 10 or higher indicates severe stroke symptoms, we excluded participants with a score of MORE than 10 as we judged that they may have difficulty understanding the questionnaire. “ Please, correct the information.
Lines 107-108: “Since an NIHSS score of 10 or higher indicates severe stroke symptoms…” However, we can see in literature:
Score | Stroke severity |
---|---|
0 | No stroke symptoms |
1–4 | Minor stroke |
5–15 | Moderate stroke |
16–20 | Moderate to severe stroke |
21–42 | Severe stroke |
Additionally, considering NIHSS, a total score more than 10 could not include alteration of the level of consciousness. Please, think about this.
2- “Gratitude”
“The participants received a mouthwash worth 3,000 as a token of gratitude for responding to the questionnaire.” 3,000 won? Please, include information in dollar.
Author Response
Thank you for reviewing this study thoroughly. Your comments on the revisions are attached.
